# Host genetic selection for cold tolerance shapes microbiome composition and modulates its response to temperature

**Fotini Kokou[1,2], Goor Sasson[1], Tali Nitzan[2], Adi Doron-Faigenboim[3], Sheenan Harpaz[2], Avner Cnaani[2], Itzhak Mizrahi[1]***

[1]Department of Life Sciences and the National Institute for Biotechnology in the Negev, Ben-Gurion University of the Negev, Beer-Sheva, Israel; [2]Department of Poultry and Aquaculture, Institute of Animal Sciences, Agricultural Research Organization, Rishon LeZion, Israel; [3]Department of Vegetable and Field Crops, Institute of Plant Science, Agricultural Research Organization, Rishon LeZion, Israel

**Abstract** The hologenome concept proposes that microbes and their host organism are an independent unit of selection. Motivated by this concept, we hypothesized that thermal acclimation in poikilothermic organisms, owing to their inability to maintain their body temperature, is connected to their microbiome composition. To test this hypothesis, we used a unique experimental setup with a transgenerational selective breeding scheme for cold tolerance in tropical tilapias. We tested the effects of the selection on the gut microbiome and on host transcriptomic response. Interestingly, we found that host genetic selection for thermal tolerance shapes the microbiome composition and its response to cold. The microbiomes of cold-resistant fish showed higher resilience to temperature changes, indicating that the microbiome is shaped by its host's selection. These findings are consistent with the hologenome concept and highlight the connection between the host and its microbiome's response to the environment.
DOI: https://doi.org/10.7554/eLife.36398.001

*For correspondence:
imizrahi@bgu.ac.il

**Competing interests:** The authors declare that no competing interests exist.

## Introduction

Cold temperature is an environmental challenge that greatly affects metabolic and physiological processes. Thus, adaptations to temperature fluctuations are expected to be found across all organisms. When exposed to cold temperatures, animals undergo remarkable physiological adjustments in order to maintain homeostasis. Such adjustments may occur through genetic and/or non-genetic mechanisms that increase their fitness. Recent work in mice revealed that the gut microbiome facilitates key adaptations during cold exposure by promoting energy demand-driven regulation (*Chevalier et al., 2015*; *Rosenberg and Zilber-Rosenberg, 2018*). This is not surprising, as many studies have already proven the vital importance of gut microbial communities for host survival, homeostasis, development and functioning (*McFall-Ngai et al., 2013*; *Shabat et al., 2016*). In fact, the universality of host–microbe associations, either transient or tight, inspired the hologenome concept (*Bordenstein and Theis, 2015*; *Brucker and Bordenstein, 2013*; *Rosenberg and Zilber-Rosenberg, 2013*; *Rosenberg and Zilber-Rosenberg, 2016*; *Rosenberg and Zilber-Rosenberg, 2018*; *Theis et al., 2016*), which proposes that within the holobiont, units at different levels, such as genes, chromosomes or biont combinations (i.e., host and microbes), are subject to selection or neutrality. Within this concept, host–microbe interactions have an important role in the host's physiology, whilst microbiome composition may be affected by host selection. Taking this into account, we hypothesized that host selection may facilitate changes in the host-associated microbial species and in their response to environmental selection pressure.

**eLife digest** Animals and plants host diverse microbial communities that are vital for their survival. In fact, the host organisms and their associated 'microbiome' are so closely linked that they are often described as a single entity: the holobiont unit. This suggests that when the host adapts to cope with stressful conditions, similar changes should also occur in its microbiome.

Fish are unable to maintain a stable body temperature and can be greatly affected by temperature fluctuations. Some fish are better able to tolerate cold conditions than others, but it was not known if their gut microbes are similarly affected by changes in temperature.

To investigate, Kokou et al. selectively bred tropical blue tilapia to create families of fish that could either tolerate the cold well, or that were highly sensitive to the cold. The gut microbiomes of cold-resistant fish were different from the cold-sensitive ones, even though the fish lived in the same tank. Moreover, the gut microbiomes of the cold-tolerant fish showed higher resilience to temperature changes than the microbes in the guts of the cold-sensitive fish.

It remains to be determined whether the response of the microbiome directly affects how its host fish responds to temperature changes. However, the results presented by Kokou et al. show that there are links between how the host and its microbes adapt to environmental stress. As well as helping us to understand how holobionts evolved, this knowledge could also potentially be applied broadly in clinical sciences or agriculture, for example to select for efficient crops.
DOI: https://doi.org/10.7554/eLife.36398.002

Poikilothermic organisms, such as fish, must develop strategies to maintain homeostasis within diverse temperature gradients because environmental temperature changes would otherwise disrupt homeostasis and cause deleterious effects on vital physiological functions (*Guschina and Harwood, 2006*; *Vinagre et al., 2012*). The physiological response to stress may vary among populations or individuals and can be affected by various factors (*Nitzan et al., 2016*; *Somero, 2010*). Microbial communities in aquatic environments have also been reported to be affected by abiotic factors that act as a habitat-filtering force, such as temperature (*Fuhrman et al., 2008*). In poikilothermic organisms, the gut microbiome experiences temperature fluctuations that do not exist in homeotherms. There is evidence of a temperature effect on the microbial dynamics associated with invertebrate species (*Beleneva and Zhukova, 2009*; *Carlos et al., 2013*; *Erwin et al., 2012*; *Lokmer and Mathias Wegner, 2015*; *Mahalaxmi et al., 2013*; *Preheim et al., 2011*; *Zurel et al., 2011*), but such information is largely missing for aquatic poikilothermic vertebrates.

Taking the hologenome concept into account, we asked whether host transgenerational selection for thermal tolerance, which is instrumental to poikilothermic organisms' fitness, involves the host–gut microbiome axis. More specifically, how does microbiome composition respond to temperature changes and does host tolerance shape this response? Here, we hypothesized that the thermal tolerance of poikilothermic vertebrates affects their associated gut microbial species, as well as the response of these microbes to temperature alterations. We focused specifically on the association of changes in the microbiome with the transgenerational low-temperature response in tropical fish, such as tilapia species. The growth of these fish species is highly affected by temperature; they are subject to growth inhibition and mortality when exposed to low temperatures (*Cnaani et al., 2000*). However, recent work revealed transgenerational inheritance of cold tolerance in these species (*Nitzan et al., 2016*). Therefore, we used a unique setup, composed of 66 fish families that are part of a transgenerational selective-breeding scheme for enhanced cold tolerance (*Figure 1*). In this setup, we chose fish progeny from families with the most extreme phenotypes, selected on the basis of the survival rate of their siblings at low temperatures and characterized as cold-resistant or cold-sensitive. Hence, these fish progenies had never experienced low temperature exposure before they were challenged. We used this setup to understand how selection on host thermal tolerance corresponds to microbiome shifts. Our results suggest that host selection for cold tolerance is followed by changes in the gut microbiome, which are manifested by higher resilience to temperature shifts and which may be a key factor in orchestrating cold acclimation in poikilothermic animals.

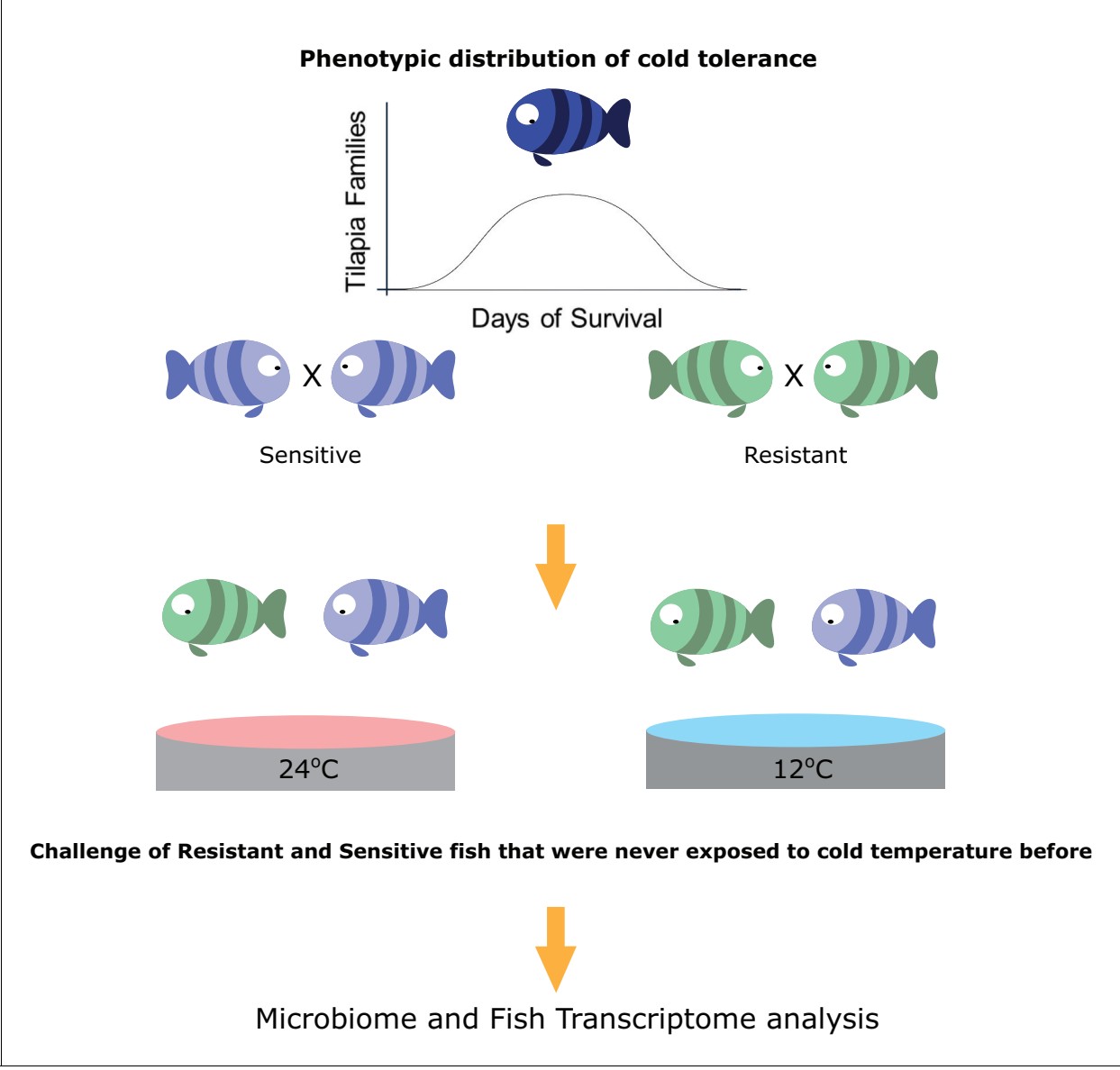

**Figure 1.** Selection for cold-adapted individuals on the basis of a breeding scheme. The fish parental lines used in this study originated from an Israeli strain of *Oreochromis aureus* which was subjected to an ongoing selective breeding program for optimal harvest weight at the Dor Aquaculture Research Station (*Zak et al., 2014*). Individuals were characterized as cold-resistant or cold-sensitive on the basis of days of survival in low temperatures (<16℃) over the generations. The three most extreme phenotypes (resistant and sensitive) based on their performance at low temperatures were selected and their siblings were transferred to the Agricultural Research Organization (ARO) – Volcani center. These animals were challenged, for the first time, by decreasing water temperature from 24℃ to 12℃ at a rate of 1 ℃/day and then held at 12℃ for 2 days before sampling the intestine (for 16S rRNA gene sequencing) and liver (for RNA sequencing) from 7 fish per family (21 fish per tolerance group in each temperature [24℃, control, and 12℃]).
DOI: https://doi.org/10.7554/eLife.36398.003

The following figure supplement is available for figure 1:

**Figure supplement 1.** Distribution of survival time (in days) in temperatures below 16℃ of the different tilapiine families and selection (highlighted) of individuals originating from the two most extreme maternal phenotypes (resistant in green and sensitive in purple).
DOI: https://doi.org/10.7554/eLife.36398.004

## Results

### Temperature is a major factor shaping the fish gut microbiome

We analyzed microbiome populations with respect to two potential habitat-filtering forces: temperature and host thermal tolerance through genetic selection. Temperature is known to affect microbial composition in aquatic habitats and to act as a habitat filter (*Fuhrman et al., 2008*). There is some limited information on the effect of temperature on the gut microbiome, showing compositional changes in homeotherms (*Chevalier et al., 2015*). However, such information is largely missing in poikilothermic vertebrates, in which physiology is strongly influenced by environmental temperature.

To explore whether the thermal tolerance of poikilothermic vertebrates affects their associated gut microbial species and their response to temperature alterations, we used the blue tilapia, *Oreochromis aureus*, as our experimental model. The optimal temperatures for this species, similar to those for other tilapiine species, are 24–28°C (*Trewavas, 1983*), but this species is one of the most cold-tolerant among this group of fish (*Cnaani et al., 2000*). An ongoing selective-breeding process, which has been running for the past few years in Israel, is aimed at improving growth and cold-tolerance traits in this species (*Zak et al., 2014*). In a recent study, we found that the fish's tolerance to temperatures below 16°C is transgenerational and potentially connected to maternal effects (*Nitzan et al., 2016*). Thus, the present study focused on understanding host–microbiome interactions in relation to temperature stress and cold acclimation.

To evaluate the responses of the fish that are resistant and sensitive to cold exposure, we challenged a total of 84 fish, sampled at 24°C or 12°C, originating from 3 tilapia families with high tolerance and 3 with low tolerance to cold temperatures (7 fish per family). For our experiments, we chose fish originating from mothers that had the most extreme phenotypes, selected on the basis of their sibling's survival rate at low temperature, as determined by previous cold-challenge trials (*Figure 1*; *Figure 1—figure supplement 1*). All of the fish within each generation were kept in the same tank to eliminate confounding effects. The test groups, consisting of 42 fish each from resistant and sensitive families, were challenged by temperature reduction to 12°C over a period of 2 weeks and compared to a control group, consisting of 42 fish from the same families kept at 24°C. At the end of the temperature-challenge trial, we analyzed the fish gut microbiomes from the test and control groups using 16S rRNA gene sequencing, and we also looked at the host's response using liver transcriptomic analysis (*Figure 1*).

Linear mixed-effects model analysis indicated that temperature was the major factor shaping microbial community diversity and richness within the fish gut (*Table 1*; $F_{Shannon}$ = 46.43, $F_{Richness}$ = 45.79, p = 0.001). Indeed, we found a dramatic decrease (*Figure 2Ai*; Wilcoxon t-test, two-sided, 95% confidence interval (CI), p = 7.2E-02) in microbial diversity (Shannon index H') and richness (*Figure 2—figure supplement 1*; Wilcoxon t-test, two-sided, 95% CI, p = 2.7E-08; *Figure 2—figure supplement 2*, rarefaction curves) when both groups were exposed to cold temperature. This decrease was in agreement with the low-temperature decrease in the 16S gene copy numbers (real-time PCR; *Figure 2—figure supplement 3*). Furthermore, the microbial communities in the fish that were subjected to cold temperatures exhibited lower variability in richness between samples, whereas greater individual variation was observed in the high temperature group (*Figure 2—figure supplement 1i*), suggesting that low-temperature conditions are very selective and potentially constrain microbial communities, decreasing their inter-individual variation.

Microbial community structure (β-diversity) was also dramatically affected by cold exposure, being the major factor associated with increasing similarity between individuals in the communities (*Table 2*; $F_{Temperature}$ = 6.5072, p = 0.002), after gut physiology effects ($F_{Part}$ = 28.8936, p = 0.001). Orders from the *Proteobacteria* phylum, and more specifically *Vibrionales* and *Alteromonadales*, were enriched at 12°C (*Figure 2B*; *Supplementary file 1*, Table S1; *Figure 2—figure supplement 4*), with prior studies reporting these taxa's potential for survival in cold temperatures (*Lauro et al., 2011*; *Math et al., 2012*; *Raymond-Bouchard and Whyte, 2017*). Overall, however, as microbial diversity and richness decreased (*Figure 2Ai*; *Figure 2—figure supplement 1i*), several microbial phyla, such as *Planctomycetes*, *Bacteroidetes* and *Verrucomicrobia*, were significantly depleted after cold-exposure (*Figure 2B*; *Figure 2—figure supplement 5*; *Supplementary file 1*, Table S1). These findings indicate that in poikilothermic vertebrates such as the blue tilapia, gut microbial communities are strongly affected by environmental temperature.

**Table 1.** Linear mixed-effects model by restricted maximum likelihood (REML) for temperature and tolerance on richness and diversity.

| | d.f. | F | P-value | Significant contrasts |
|---|---|---|---|---|
| *Main effects – Richness* | | | | |
| AIC = 1559.913, BIC = 1578.438, logLik = −773.956 | | | | |
| Temperature | 149 | 45.79 | <0.001* | Warm higher than cold |
| Tolerance – Sensitive | 149 | 5.42 | 0.0213* | Sensitive higher than resistant |
| *Main effects – Diversity (Shannon index H′)* | | | | |
| AIC = 378.6602, BIC = 397.1857, logLik = −183.3301 | | | | |
| Temperature | 149 | 46.43 | <0.001* | Warm higher than cold |
| Tolerance – Sensitive | 149 | 5.37 | 0.0218* | Sensitive higher than resistant |

*Statistical significance at p < 0.05.

d.f., degrees of freedom; AIC, Akaike information criterion; BIC, Bayesian information criterion; logLik, log of likelihood.

DOI: https://doi.org/10.7554/eLife.36398.014

## Host genetic selection for thermal tolerance primes for a specific microbiome composition and buffers the temperature effect on it

After observing a drastic decrease in gut microbial diversity during cold exposure (*Figure 2Ai*), we sought to evaluate how the host's thermal tolerance affects microbiome composition. Moreover, if host thermal tolerance is connected to cold temperature acclimation, then we would expect the microbiome response to relate to different tolerance phenotypes. We tested this hypothesis by evaluating the response of the cold-sensitive and cold-resistant selected families to the changing environmental conditions. We compared the gut microbial compositions of the sensitive and resistant families and found that host thermal tolerance has a significant effect on gut microbial diversity (*Table 2*; $F_{Tolerance}$ = 2.2269, p = 0.04). Furthermore, we also observed a significant interaction between host thermal tolerance and temperature, indicating a different microbiome response to the lower temperature in the resistant vs. sensitive hosts ($F_{TxT}$ = 3.6113, p = 0.012). Interestingly, when fish from both groups were kept under their optimal temperature conditions of 24°C, the sensitive fish exhibited higher microbial diversity (Shannon H′) than the resistant ones, as well as higher individual variability (*Figure 2Aii*; Wilcoxon t-test, two-sided, 95% CI, p = 0.048). Indeed, when we compared the β-diversity of the two groups under these optimal conditions, we found that cold-resistant fish had a higher within-group microbiome similarity than the sensitive fish (*Figure 2Ci*; non-parametric Bonferroni-corrected p = 0.01, using 1000 Monte Carlo permutations), potentially because of the selection process to which resistant fish have been subjected with regard to the cold-resistance trait, thus showing host control of microbiome composition.

We further aimed to gain better insight into the microbiome composition of these host groups. We first compared the microbiome composition of cold-resistant and cold-sensitive hosts under each temperature condition; more specifically, we asked whether microbiomes of similar (Shannon H′) diversity also share the same microbiome compositions. To answer this, we stratified our data and examined individuals from each host group with similar Shannon H′ diversity (individuals from each group with the 10 highest or 10 lowest Shannon H′ diversity values; *Supplementary file 1*, Table S2). We found that the individuals from both high- and low-diversity microbiomes (Principal coordinate analysis; *Figure 2—figure supplement 6*) clustered significantly according to host group (Permanova analysis; *Supplementary file 1*, Tables S3 and S4), suggesting that host genetic background (genetic selection for cold tolerance) had a strong effect on shaping the microbiome composition.

Our next step was to explore the microbial taxa that are associated with genetic background effects. Indicator species analysis (which identifies habitat-associated species on the basis of their

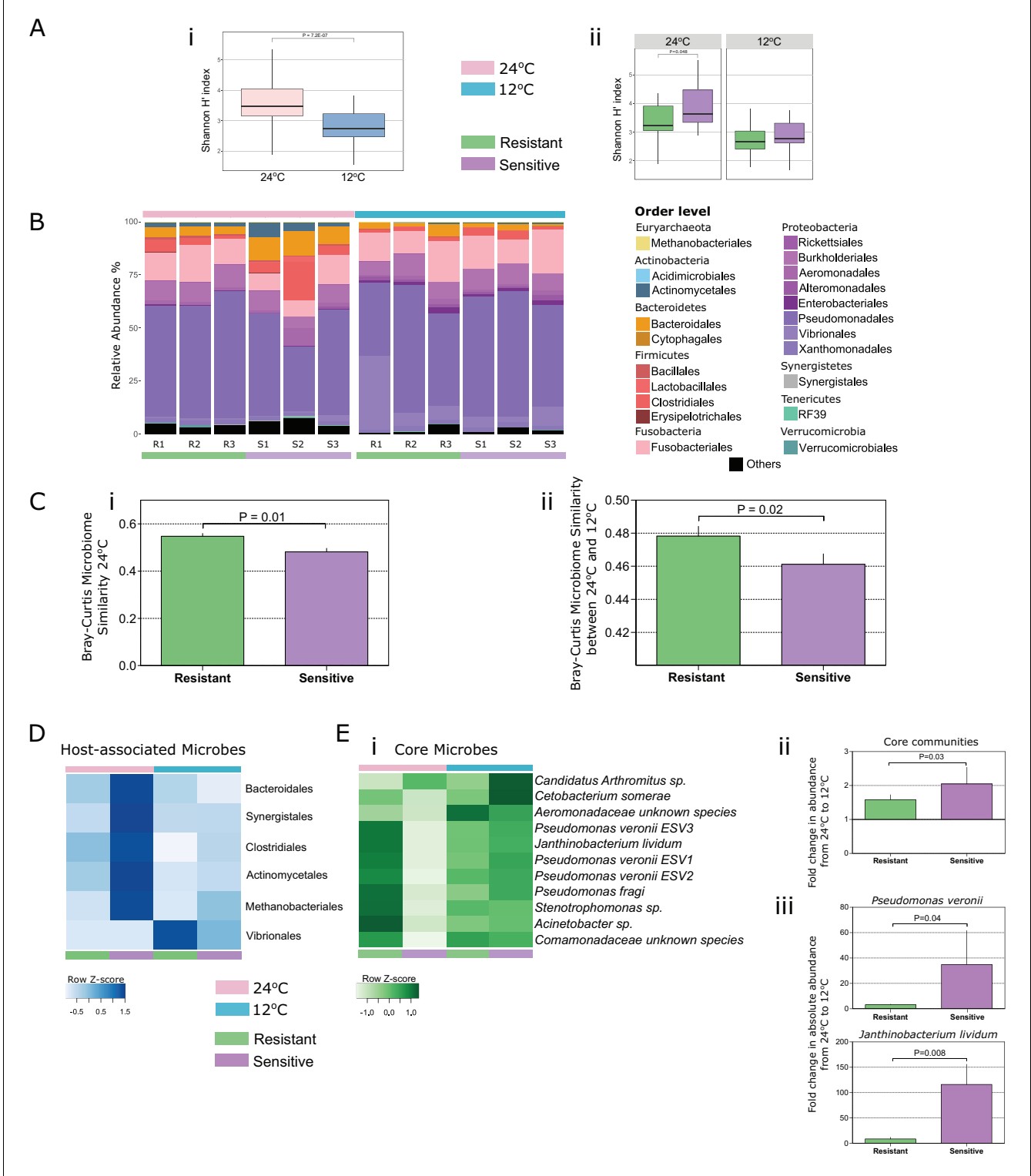

**Figure 2.** Both temperature and host thermal tolerance have a major effect on shaping microbial communities in the fish gut. (**A**) (i) Shannon H' diversity was decreased (Wilcoxon t-test, two-sided, 95% CI), when fish from both groups were exposed to cold conditions (12°C, blue). (ii) However, in control-warm conditions (24°C, pink) sensitive fish (purple) had a greater diversity of species in their gut microbiomes compared to the resistant fish (green) (Wilcoxon t-test, two-sided, 95% CI). (**B**) The relative abundance of various orders of microbes in the gut was affected when both resistant and sensitive fish were moved from warm to cold conditions. (**C**) Comparison of the β-diversity (Bray Curtis) within each of the host groups in (i) warm conditions showed a greater similarity among the resistant individuals compared to that among the sensitive fish, while (ii) comparison in relation to

*Figure 2 continued on next page*

*Figure 2 continued*

temperature change showed that after exposure to the stressful low-temperature conditions, the microbiomes of cold-resistant fish families were less affected than those of sensitive ones (non-parametric Bonferroni-corrected, using 1000 Monte Carlo permutations). (D) Host-associated species (from indicator species analysis; see 'Materials and methods' section) include several orders that were mostly enriched in either the resistant or the sensitive hosts. (E) (i) Microbial species that were present in >50% of the individuals in both warm and cold conditions (core communities) responded with an increase in their abundance after exposure to cold conditions in the sensitive group, and remained stable in the resistant fish (fold change in abundance from warm to cold shown in (ii)). (iii) These results were also validated by quantitative PCR analysis of the most abundant core species (*Pseudomonas veronii* and *Janthinobacterium lividum*). In the box plots, horizontal line in the box represents the median and whiskers indicate the lowest and highest point within 1.5 interquartile ranges of the lower or upper quartile, respectively. Bar plots show the mean copy number per group and whiskers the standard error of the mean.

DOI: https://doi.org/10.7554/eLife.36398.005

The following figure supplements are available for figure 2:

**Figure supplement 1.** Temperature significantly decreased richness in both resistant and sensitive fish groups.
DOI: https://doi.org/10.7554/eLife.36398.006

**Figure supplement 2.** Rarefaction curves for sequence depth in the different temperatures.
DOI: https://doi.org/10.7554/eLife.36398.007

**Figure supplement 3.** Total bacterial 16S gene copy number within each group at 24°C (pink) and 12°C (blue) in the (i) anterior and (ii) posterior gut of resistant and sensitive fish.
DOI: https://doi.org/10.7554/eLife.36398.008

**Figure supplement 4.** Fold-change in the taxa abundance at different taxonomic levels ((A) phylum, (b) class, (c) order, (d) family, (e) genus and (f) species) from 24°C to 12°C within the guts of the resistant and sensitive host groups.
DOI: https://doi.org/10.7554/eLife.36398.009

**Figure supplement 5.** Relative abundance of the most abundant microbial taxa within the posterior gut of fish in the two temperature conditions, at the (i) family, (ii) genus and (iii) exact sequence variant (ESV) levels.
DOI: https://doi.org/10.7554/eLife.36398.010

**Figure supplement 6.** Principal coordinate analysis (Bray-Curtis similarity metric) of selected individuals with (i) high or (ii) low Shannon H' diversity index (see *Supplementary file 1*, Table S2) in (i) warm (red) and (ii) cold (blue) conditions, respectively.
DOI: https://doi.org/10.7554/eLife.36398.011

**Figure supplement 7.** Relative abundance of host-associated (indicator) species that were significantly enriched in low and high diversity microbiomes (see *Supplementary file 1*, Table S5 for P-values), in (i) warm and (ii) cold conditions in resistant and sensitive fish hosts.
DOI: https://doi.org/10.7554/eLife.36398.012

**Figure supplement 8.** Indicator species analysis (see 'Materials and methods' section) revealed several exact sequence variants that were specific to either the resistant or sensitive fish posterior gut.
DOI: https://doi.org/10.7554/eLife.36398.013

fidelity and relative abundance in different environments; see 'Materials and methods') revealed several microbial taxa that were significantly associated with host genetic background and had been described previously as naturally occurring strains in the tilapia gut (*Giatsis et al., 2016*;

**Table 2.** Permanova (Adonis) results for experimental communities based on Bray–Curtis distances.

| | d.f. | SS | MS | PseudoF | R² | P-value |
|---|---|---|---|---|---|---|
| *Main Effects* | | | | | | |
| Tolerance | 1 | 0.3112 | 0.3112 | 2.2269 | 0.01102 | 0.04* |
| Temperature | 1 | 0.9093 | 0.9093 | 6.5072 | 0.03220 | 0.002** |
| Family | 1 | 0.2601 | 0.2601 | 1.8611 | 0.00921 | 0.110 |
| Part | 1 | 4.0374 | 4.0374 | 28.8936 | 0.14297 | 0.001** |
| *Interaction terms* | | | | | | |
| Tolerance x temperature | 1 | 0.5046 | 0.5046 | 3.6113 | 0.01787 | 0.012* |
| Residuals | 159 | 22.2176 | 0.1397 | | 0.78674 | |
| Total | 164 | 28.2401 | | | 1.00000 | |

*, **Statistical significance at p < 0.05 and 0.01, respectively. Permutations n = 999.
d.f., degrees of freedom; SS, sum of squares; MS, mean sum of squares.
DOI: https://doi.org/10.7554/eLife.36398.015

*Haygood and Jha, 2016*) (*Supplementary file 1*, Tables S5 and S6; *Figure 2—figure supplements 7–8*). Taxa including microbial species such as *Cetobacterium somerae* (*Tsuchiya et al., 2008*) and species of the order *Vibrionales* (i.e., *Vibrio cholerae*; *Figure 2—figure supplement 5F*), which were also described as microbes surviving in environments with a broad temperature range (*Raymond-Bouchard and Whyte, 2017*; *Townsley et al., 2016*), were enriched in the resistant fish gut (*Figure 2D*; *Figure 2—figure supplement 8*; *Supplementary file 1*, Tables S5 and S6). On the other hand, a higher diversity of taxa, including *Prevotella* sp., *Streptococcus luteciae* and Bacteroidetes species, as well as species from the families *Christensenellaceae, Succinivibrionaceae* and *Clostridiaceae*,was enriched in the sensitive fish gut (*Figure 2D*; *Supplementary file 1*, Tables S5 and S6). Altogether, our results indicate that host genetic background shapes microbiome composition by selecting for specific microbes, with some of them potentially carrying fitness traits for low-temperature tolerance.

As host fish were specifically selected according to their thermal tolerance, and as our findings indicated that temperature and host genetic background affect the fish gut microbiome, we further asked whether the microbiome's response to temperature stress is also affected by its host's tolerance. When exposed to the cold temperature of 12°C, both resistant and sensitive hosts exhibited a decrease in their microbiome diversity and richness (*Figure 2A*; *Figure 2—figure supplement 1*), as well as changes in microbiome composition (*Figure 2B*; *Figure 2—figure supplements 4–8*). When we compared the β-diversity within each of these host groups in relation to changing temperature, however, we found that after exposure to the stressful low-temperature conditions, the microbiomes of cold-resistant fish families were less affected than those of sensitive ones (*Figure 2Cii*; non-parametric Bonferroni-corrected p = 0.02, using 1000 Monte Carlo permutations). More specifically, when we compared the similarity of the microbial compositions between fish at 24°C and 12°C, we found it to be significantly greater in the resistant fish families than in the sensitive ones, showing a stronger effect of temperature change in the sensitive fish (*Figure 2Cii*).

We next asked whether the microbiome's differential response to temperature in the two host groups also applies to microbes that are shared between the two host groups. We assessed microbiome resilience by looking at 11 selected taxa that were present in >50% of the individuals from each temperature environment (the 'core microbiome'). After renormalizing the core microbes' abundance (expressed as percentage within the core microbiome community; *Figure 2E*), we evaluated the fold changes in the abundance of these taxa upon change from 24°C to 12°C in both the cold-resistant and cold-sensitive fish. Similar to the overall microbiome response (*Figure 2Cii*), the fold change in the relative abundance of the core microbiome in resistant fish was significantly lower than that in sensitive ones (*Figure 2Eii*). We further designed specific primers to measure the absolute copy numbers of the two most abundant and prevalent (>95% of the individuals) core taxa (*Pseudomonas veronii* and *Janthinobacterium lividum*) in both host groups and quantified their abundance by quantitative PCR (see 'Materials and methods'). Our results were in agreement with the sequencing data (*Figure 2Ei*, ii), showing a significantly less pronounced temperature effect on the abundance of the core microbes in the gut of the resistant fish compared to that in the gut of the sensitive ones (*Figure 2Eiii*). These results thus supported our hypothesis that host thermal tolerance modulates the microbiome's response to temperature changes.

## Microbial functional diversity corresponds to cold stress and host acclimation

Since we found that microbiome composition and response to temperature changes are connected to the host's thermal tolerance, we next evaluated the microbial community composition and attributes in relation to temperature and host cold acclimation. We utilized the linear discriminant analysis (LDA) effect size (LEfSe) tool (*Segata et al., 2011*) and the PICRUSt (Phylogenetic Investigation of Communities by Reconstruction of Unobserved States) tool (*Langille et al., 2013*). During cold exposure, we found a significant change (enrichment or depletion) in several pathways in all host microbiomes, resistant and sensitive. More specifically, a decrease in the abundance of genes belonging to metabolic pathways was observed, whereas genes in pathways related to cellular processes and signaling increased significantly in abundance (*Figure 3A*). In line with the changes in microbial composition, the overall fold change in microbial coding capacity between warm and cold temperature was significantly greater in the cold-sensitive fish (*Figure 3B*; p < 0.0001, Wilcoxon rank-sum test, two-sided), showing a more pronounced response in this group. Specifically,

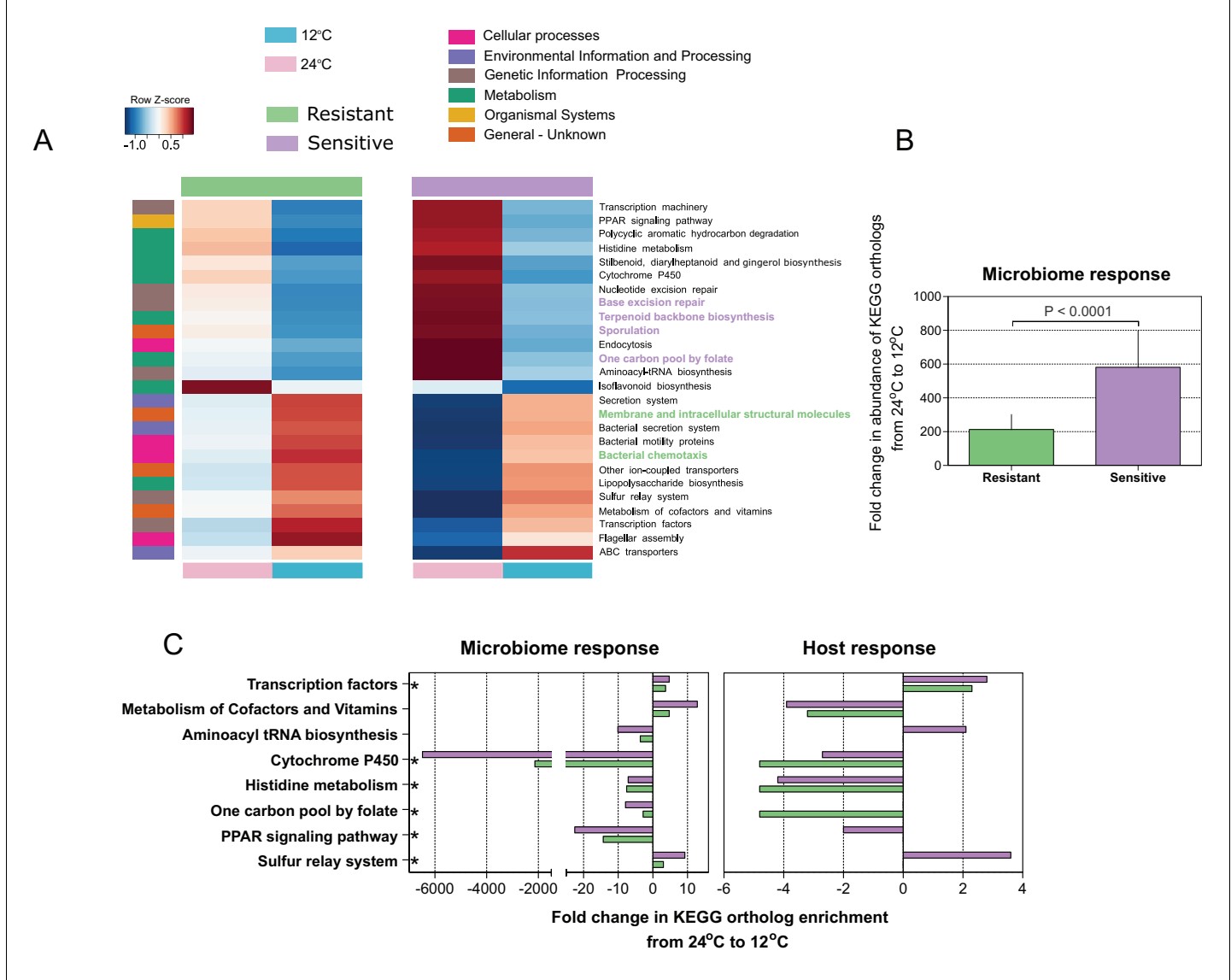

**Figure 3.** Microbial functional diversity corresponds to cold stress and the host's acclimation. (A) Pathway enrichment analysis using PICRUSt-predicted KEGG (Kyoto Encyclopedia of Genes and Genomes) orthologs revealed a decrease in microbiome pathways related to metabolism during cold exposure, with an increase in the number of ortholog groups related to cellular processes, genetic and environmental information and processing. Different pathways were enriched in the resistant and sensitive fish, as indicated by the LEfSe analysis (see 'Materials and methods' section). (B) Differences were observed in the microbiome responses of the cold-resistant and sensitive fish, the latter showing a higher fold change from control when exposed to cold conditions (Wilcoxon rank-sum pairwise test, p < 0.0001, two sided). The bar plots show the mean fold change in abundance of KEGG orthologs in the microbial pathways after temperature change from 24°C to 12°C and whiskers represent the standard error of the mean, as indicated by PICRUSt and LEfSe analysis. (C) Functions that are shared between the microbiome and host response that were affected after cold exposure. Bar plots show fold change in the enrichment of KEGG orthologs after temperature change from 24°C to 12°C.

DOI: https://doi.org/10.7554/eLife.36398.016

The following figure supplement is available for figure 3:

**Figure supplement 1.** Pathways affected during cold exposure in the resistant and sensitive fish families.

DOI: https://doi.org/10.7554/eLife.36398.017

microbiome composition changes resulting from cold exposure seem to relate to functional shifts, which overall correspond to a stress-related response and to cell defense, in line with prior studies suggesting reduction of metabolic activity and transcription, and increases in motility, chemotaxis and membrane transport in response to cold (*Nachin et al., 2005*). Therefore, host thermal tolerance seems to contribute to and to enhance the differences in microbiome functions with respect to

thermal stress, potentially suggesting that not only microbiome composition, but also certain attributes of the microbiome may be influenced by host genetic background and response to low temperatures.

## Host–microbiome response during cold exposure indicates a more stable phenotype

To this point, our findings indicated that selection for host thermal tolerance shapes microbiome composition and response to decreased temperatures. We then asked whether the host response to the change in temperature in the resistant and sensitive families agrees with the microbiome response, as it is known that cold exposure greatly affects metabolic and physiological processes in poikilothermic vertebrates, such as fish, toward maintaining fitness and adaptation (*Schulte et al., 2011*). Specifically, we asked whether host response in the resistant fish families is more moderate than that in the sensitive families. We sequenced the transcriptome in the liver, which is an important and sensitive organ for stress and immune regulation in fish (*Kokou et al., 2016*; *Möller et al., 2014*). In agreement with our findings of a differential microbiome response to cold exposure between the resistant and sensitive fish, the transcriptomic response in the liver showed that the overall fold change in gene expression between warm and cold temperature conditions was significantly higher in the cold-sensitive fish (*Figure 4A*; $p < 0.0001$, Wilcoxon rank-sum test, one-sided), showing a more pronounced response in this group. Moreover, when we compared the similarity in the transcript presence/absence patterns of the host response between fish in 24°C and 12°C conditions, we found it to be significantly higher in the resistant families than in the sensitive ones, showing a stronger effect of temperature change in the sensitive fish (*Figure 4B*; *Figure 4—figure supplement 1*), as was also the case for the microbiome response (*Figure 3B*). Similarly, when we analysed the transcriptomic response for host-associated genes (sensitive vs. resistant; see 'Materials and methods'), the response of the resistant hosts to a change in temperature from 24°C to 12°C changed significantly less than that of the sensitive fish (*Figure 4C*; *Figure 4—figure supplement 2*).

To further test such agreement between host and microbiome responses, we compared the posterior gut microbiome structure and the liver transcriptome of resistant and sensitive fish. Specifically, we performed a Principal Component Analysis (PCA) for both the microbiome and transcriptome and we correlated the top five microbiome principal components (PCs) to the top five transcriptome PCs by means of Canonical Correlation Analysis (CCA), which aims to maximize and measure the correlation between two multivariate datasets. Interestingly, the measured correlation coefficient was significant when we compared it to correlation coefficients achieved after randomly shuffling the sample labels (*Figure 4B*; $p = 0.0044$). This analysis showed that more than 80% of the variance observed in the microbiome response to temperature could be explained by the variance observed in the transcriptomic response (*Figure 4B*; actual variance was 83% vs. permuted 44%), thus supporting a host–microbiome interaction in response to environmental changes. Notably, when we looked at the host transcriptome response to cold exposure, we found several metabolic pathways (*Figure 3C*) that were also affected in the microbiome (*Figure 3A*), with most of the changes occurring in the same direction (depletion or enrichment). Looking at the functions that are commonly shared between the host and the microbiome response to cold exposure (*Figure 3C*), we found that some of these functions, namely transcription- and cytochrome P450-related pathways (*Figure 3C*; *Figure 3—figure supplement 1*), were also reported as key cell metabolism cold-induced adaptations in homeotherms (*Shore et al., 2013*). Such findings led us hypothesize that these functions may be conserved with regard to host and gut microbiome responses to temperature changes.

## Discussion

Within the hologenome concept, host–microbe interactions are connected and make up a unit of natural selection (*Rosenberg and Zilber-Rosenberg, 2013*; *Rosenberg and Zilber-Rosenberg, 2016*; *Rosenberg and Zilber-Rosenberg, 2018*; *Theis et al., 2016*). Inspired by this concept, we hypothesized that this association between the host and its microbes may relate to the host's response to environmental changes, such as exposure to low temperature. Cold exposure led to dramatic changes in gut microbiome composition, representing a notable determinant of the microbial

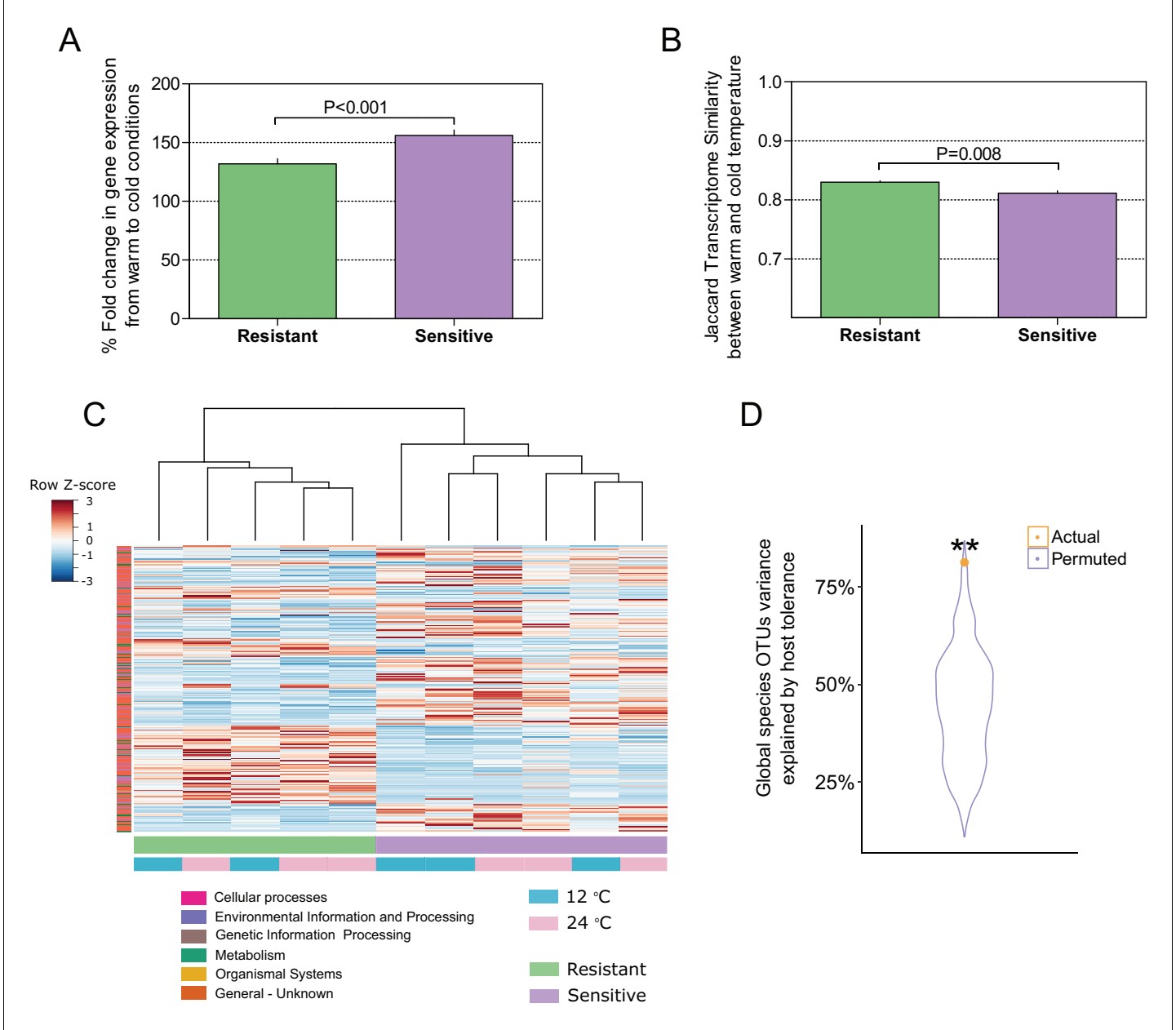

**Figure 4.** Host–microbiome response during cold exposure indicates a more stable phenotype. (**A**) Comparison of the average fold change in transcriptome expression per host group in relation to temperature change showed that after exposure to low-temperature conditions, the gene expression of cold-resistant families was less affected than those of sensitive ones (Wilcoxon rank-sum test, p < 0.0001, two-sided). (**B**) Comparison of the transcriptome β-diversity (Jaccard) in warm and cold conditions within each of the host groups showed a greater similarity among the resistant individuals than among the sensitive ones (Wilcoxon rank-run test, p = 0.008, two-sided). (**C**) Heatmap showing the expression patterns of host-associated genes as determined by indval analysis (rows) of cold-resistant and cold-sensitive hosts in warm and cold conditions (columns). Genes were annotated in KEGG (Kyoto Encyclopedia of Genes and Genomes). (**D**) Total fraction of operational taxonomic unit (OTU) variance explained by host genetics on the basis of Canonical Correlation Analysis (CCA) between the matrices of OTUs principal coordinates and transcriptomic data principal coordinates. A significant correlation was found after comparing the posterior gut microbiome structure and the liver transcriptome between the resistant and sensitive fish (Canonical Correlation Analysis, 1000 permutations, actual variance = 0.83, permuted variance = 0.44, p = 0.0044). This actual value was compared to that of 1000 random permutations, where the order of the host lines' principal coordinates was shuffled.

DOI: https://doi.org/10.7554/eLife.36398.018

The following figure supplements are available for figure 4:

**Figure supplement 1.** Non-metric multidimensional scaling (NMDS) on the liver transcriptome data showed that clusters were formed on the basis of the Jaccard metric, between the resistant and sensitive fish lines at 12°C and 24°C.

*Figure 4 continued on next page*

*Figure 4 continued*

DOI: https://doi.org/10.7554/eLife.36398.019

**Figure supplement 2.** Comparison of the host-associated gene (indval analysis) expression similarity (Jaccard) within each of the host groups between warm and cold conditions showed a higher similarity among the resistant individuals than among the sensitive ones (Wilcoxon rank-run test, p = 0.02, one-sided).

DOI: https://doi.org/10.7554/eLife.36398.020

**Figure supplement 3.** Principal coordinate analysis using the Bray Curtis metric for the selected individuals for which transcriptomic analysis was performed.

DOI: https://doi.org/10.7554/eLife.36398.021

communities, by increasing several orders of *Proteobacteria* (*Vibrionales* and *Alteromonadales*) and decreasing all other phyla (*Figure 2*). In particular, some species of the order *Alteromonadale*s have been shown to facilitate adaptations for survival under different conditions, including low temperatures (*Alteromonas* [*Math et al., 2012*]; *Shewanella* [*Abboud et al., 2005*]), whereas several *Vibrionales* species have been found to adapt to low temperatures, such as *Vibrio cholerae* (*Townsley et al., 2016*), which was enriched under our cold conditions. These changes in microbial composition also led to a substantial change in the functional profile of the gut microbial communities. Notably, our analysis revealed that pathways related to metabolism were depleted during cold exposure, in both the microbiome and host transcriptomic response (*Figure 3*; *Figure 3—figure supplement 1*), whereas KEGG (Kyoto Encyclopedia of Genes and Genomes) orthologs related to cell communication, membrane transport and structure, signaling and cell motility were enriched, indicating a potential overall stress response (*Nachin et al., 2005*; *Tsimring et al., 1995*) (*Figure 3*). Interestingly, such changes have also been associated with ecological fitness traits (i.e., membrane transport, signal-transduction genes, tRNA diversity, carbon metabolism) (*Math et al., 2012*) that assist in microbial survival strategies during cold adaptation in some *Alteromonadales* species*Math et al., 2012*. It should be noted that the enrichment of microbial taxonomic groups that are associated with fish hosts and temperature changes, such as the *Proteobacteria* orders, potentially resulted in the enrichment of functions associated with these taxa, while not necessarily being connected to cold tolerance. However, we did find several of these functions that were not only shared between the host and the microbiome but also similarly affected in the host and microbiome after cold exposure (*Figure 3C*; *Figure 3—figure supplement 1B*). Some of the functions were actually previously reported as key cell metabolism cold-induced adaptations in mice (i.e. downregulation of peroxisome proliferator-activated receptor (PPAR) signaling and cytochrome-450-related pathways; [*Shore et al., 2013*]), thus suggesting a potential conserved response to low temperatures.

Motivated by the hologenome concept, we deduced that the microbiome response to low temperatures would also be associated with the host response and would be related to host thermal tolerance. We used two different groups of tilapias originating from mothers with different tolerance to low water temperatures (*Nitzan et al., 2016*). Interestingly, we found agreement between transcriptomic and microbiome responses to cold exposure, marking a host–microbiome interaction that potentially leads to higher microbiome stability and lower physiological stress in the cold-resistant fish (*Figure 4*). As these animals originate from different families (crosses) and not from the same population, and were kept under the same experimental conditions (same tanks) across generations, we conclude that this effect originates from differences in the genetic background, thermal tolerance and acclimation of the fish.

The decrease in microbiome richness and diversity with cold exposure in both resistant and sensitive fish could be explained in terms of selective pressure (*Figure 2A*). As these fish are poikilothermic organisms, their gut microbiome is directly exposed to water temperature. Warmer temperatures, which are optimal for tilapias, are more permissive and can lead to greater heterogeneity and richness of microbiome composition (*Erwin et al., 2012*)(*Figure 2A,B*). Alternatively, a decrease in temperature imposes a selective pressure that reduces microbial diversity, as we observed in both resistant and sensitive fish (*Chevalier et al., 2015*; *Lokmer and Mathias Wegner, 2015*). Intriguingly, our findings show host control of microbiome composition and response to temperature differences. Under warm conditions, we found compositional microbiome differences between the two host groups (*Figure 2B,C,D*), highlighting the importance of host genetic

background for microbiome composition. Nonetheless, in the transition to cold temperature, these differences in the microbiome responses in sensitive and resistant fish were manifested at the taxonomic and functional levels (*Figure 2B,D,E*), through enrichment of taxa with potential ecological fitness traits for cold adaptation or by affecting the cold sensitivity of the core microbial communities to temperature, and thus illustrating that the pre-exposure potential of the gut microbiome is due to host selection.

Cold exposure greatly affects metabolic and physiological processes in fish, which undergo several acclimation processes to maintain their fitness. The transcriptomic or physiological responses may include changes in enzymatic activity and efficiency, membrane permeability, and metabolic rate through mitochondrial energy production (*Battersby and Moyes, 1998*; *Itoi et al., 2003*), and may vary with the life history or genetic background of the organism. Transcriptomic responses in the blue tilapia liver were in line with the microbiome response to temperature, both in the size of the effect in resistant and sensitive fish (*Figure 4*; *Figure 4—figure supplements 1–2*) and in its nature, manifested as changes in functions related to metabolism and signaling (*Figure 3—figure supplement 1*). Resistant fish seemed to be under lower levels of physiological stress than sensitive fish, and therefore had lower temperature-related metabolic demands. These results could potentially be connected to lower energetic demands for maintaining homeostasis. In fact, it was previously shown that these resistant fish have lower mitochondrial gene expression than their sensitive counterparts (*Nitzan et al., 2016*). Agreement between host transcriptomic and microbiome response, showing higher stability in the resistant animals compared to the sensitive ones (*Figure 4*; *Figure 4—figure supplements 1–2*), supports our hypothesis of host–microbe associations that potentially modulate the acclimation process.

## Conclusions

Our insight into the microbial dynamics of the blue tilapia gut microbiome in response to environmental conditions shows that temperature, host genetic background and tolerance to thermal stress are major determinants of community structure and dynamics. Microbial community diversity and richness severely decreased during cold exposure; but microbial response was related to host thermal tolerance, which affected microbiome stability (*Figure 5*). These results indicate that host cold tolerance shapes not only gut microbiome composition but also the sensitivity of core microbial communities to temperature, and that host cold tolerance modulates the thermal acclimation of the gut microbiome. These finding are consistent with the hologenome concept, which proposes that associations exist between the microbes and their hosts, suggesting that together they consist a unit of natural selection. Thus, taking this concept into account, we highlight potential connections between host acclimation and its microbiome response to environmental stress.

## Materials and methods

### Experimental design and sampling

Three sensitive and three resistant families of blue tilapia (*Nitzan et al., 2016*), selected for different survival rates during cold exposure (*Figure 1—figure supplement 1*), were raised at the Dor Aquaculture Research Station, Israel, and transferred to a climate-controlled room in the Institute of Animal Science, Agricultural Research Organization (Rishon LeZion, Israel). Progenies of each mother are termed family in our dataset (we used 6 families in total - 3 resistant and 3 sensitive). Seven fish from each family were cold-challenged by gradual temperature decrease from 24°C to 12°C at a rate of 1 °C/day and then held at 12°C for 2 days, while their full-siblings were maintained at 24°C. No mortality was observed during this period. Liver tissue from three fish per family in each treatment was dissected and kept in Ambion RNAlater buffer at −20°C for transcriptome analysis. In addition, two gut compartments were sampled, the anterior and posterior intestine, from the 21 tolerant and 21 sensitive fish (7 from each family) in each treatment (in total 84 fish in cold and warm conditions; 168 samples from both anterior and posterior intestine). After dissection, the gut parts, including content and mucosa, were removed and the connective tissue cleaned; each sample was then ground, frozen and stored at −80°C for further analysis of microbiome composition. This study was approved by the Agricultural Research Organization Committee for Ethics in Using Experimental

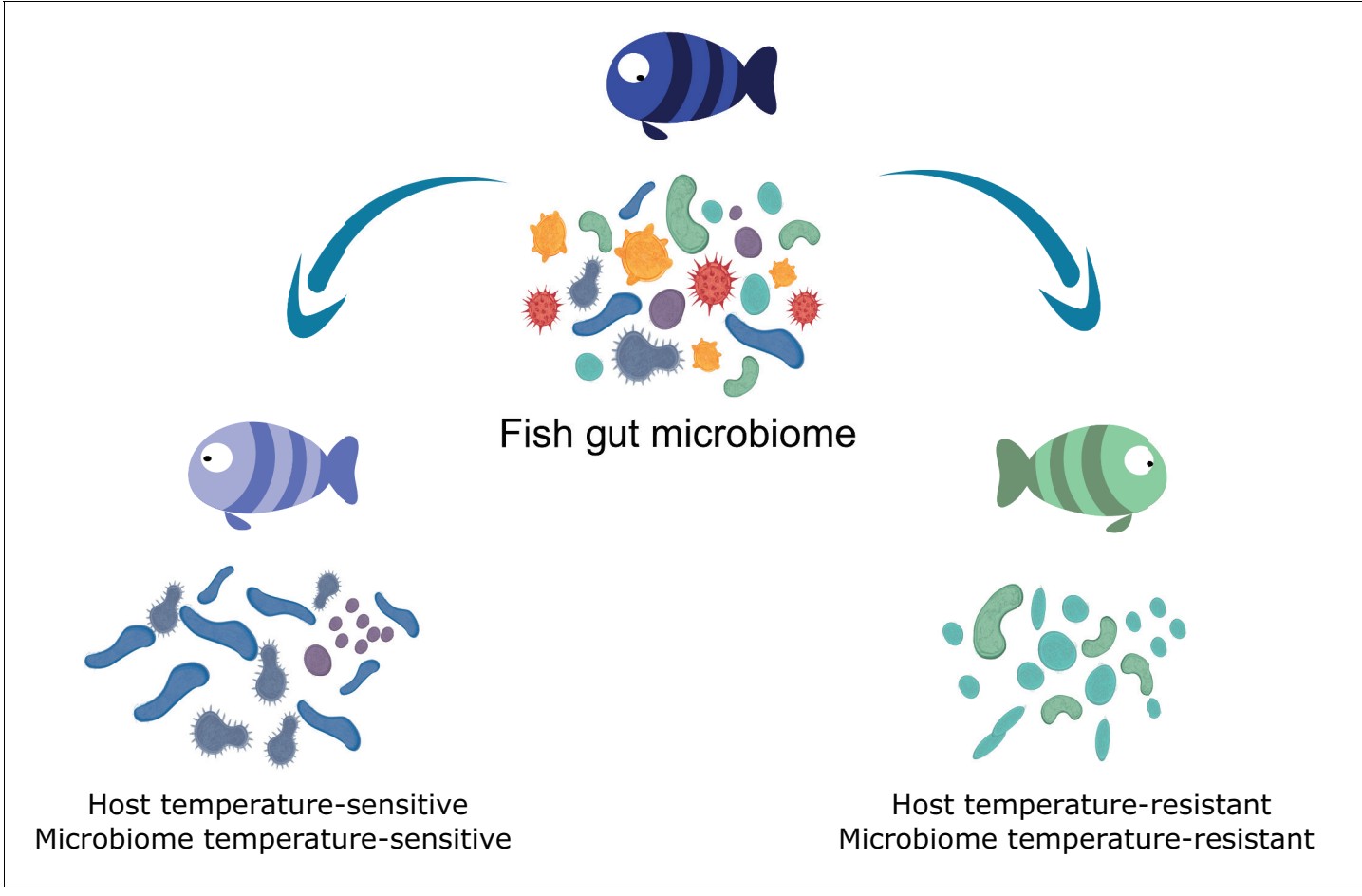

**Figure 5.** Host cold-tolerant phenotype selection is accompanied by a cold-tolerant gut microbiome.
DOI: https://doi.org/10.7554/eLife.36398.022

Animals and was carried out in compliance with the current laws governing biological research in Israel (Approval number: 146/09IL).

## DNA extraction

Bacterial DNA was isolated from gut samples using the protocol described by *Roeselers et al. (2011)* with some modifications (*Sun et al., 2013*). Excised intestines were combined in 2.0 ml screw-cap tubes with 0.5 mm and 1 mm silica beads, 400 ml 50 mM Na-phosphate buffer (pH 8.0) and 200 ml of lysis solution containing 5% w/v sodium dodecyl sulfate, 0.5 M Tris-HCl (pH 8.0) and 0.1 M NaCl. Samples were homogenized in a bead-beater for 5 min on high speed and centrifuged at 16,000 g for 5 min. The supernatant was transferred to new tubes and lysozyme (Sigma, St. Louis, MO) was added to a final concentration of 2 mg/ml followed by incubation at 42°C for 1 hr and then at 37°C for 1 hr. Following this step, the solution was sequentially extracted with TE (10 mM Tris-HCl (pH 8.0) and 1 mM EDTA), saturated phenol, phenol-chloroform (1:1 v/v), and chloroform-isoamyl alcohol (24:1 v/v). Finally, DNA in the aqueous phase was precipitated with 0.1 vol 3 M sodium acetate (pH 5.2) and 0.7 vol isopropanol. The concentration of DNA in the solution was measured using a UV-Vis Spectrophotometer (Thermo Scientific, Waltham, MA) (*Cnaani et al., 2000*) and stored at −20°C for further analysis. Only samples that resulted in a high yield of quality DNA were used for subsequent analyses.

Quantitative real-time PCR analysis was performed to measure the total number of bacteria in the anterior and posterior parts of both resistant and sensitive fish at both temperatures, through amplification of their 16S rRNA gene. Real-time PCR was performed in a 10 µl reaction mixture containing

5 µl Absolute Blue SYBR Green Master Mix (Thermo Scientific), 0.5 µl of each primer (10 mM working concentration; Forward 5′-ACTCCTACGGGAGGCAGC-3′ and Reverse 5′-GTATTACCGCGGCTGC TGGCA-3′), 3 µl nuclease-free water and 2 µl of 100 ng DNA template. Amplification involved one cycle held at 95°C for 15 min for initial denaturation and activation of the hot-start polymerase system, and then 40 cycles at 95°C for 10 s followed by annealing for 15 s at 60°C and extension at 72°C for 20 s. Bacteria were quantified using a standard curve for the 16S rRNA gene at different concentrations ($10^2$–$10^8$ copies/µl) and the results were expressed as 16S gene copy numbers per microliter.

## Sequencing of gut microbiome

Sequencing of the PCR-amplified V4 region of 16S rRNA was performed using the Next Generation system (Illumina, California, United States) (*Kanehisa and Goto, 2000*). First, amplification of the V4 region, using primers 515F (5′-GTGCCAGCMGCCGCGGTAA-3′) and 806R (in which each R contained a different 12 bp index), was performed under the following conditions: 94°C for 15 min, followed by 35 cycles of 94°C for 45 s, 50°C for 60 s and 72°C for 90 s, and a final elongation step at 72°C for 10 min. The PCR product (380 bp) was cleaned using DNA Clean and Concentrator (Zymo Research, California, United States) and the fragments containing the Illumina adaptors were quantified. Amplification involved one cycle held at 95°C for 15 min for initial denaturation and then 40 cycles at 95°C for 10 s, followed by annealing at 60°C for 20 s and extension at 72°C for 30 s. The product was quantified using a standard curve with serial DNA concentrations (0.1–10 nM). Finally, the samples were equimolarly diluted to a concentration of 0.4 nM and prepared for sequencing according to the manufacturer's instructions. An open-source software package, DADA2 (*Callahan et al., 2016*), was applied to model and correct Illumina-sequenced amplicon errors, following the tutorial suggestions (https://github.com/benjjneb/dada2). DADA2 resolves differences at the single-nucleotide level and the end product is an amplicon sequence variant table, which is a higher-resolution analog of the traditional OTU table, recording the number of times each exact sequence variant (ESV) was observed in each sample (100% sequence identity). Taxonomy was assigned using the Ribosomal Database Project Classifier (*Wang et al., 2007*) against the 16S gene reference Greengenes database (13.8 version) (*McDonald et al., 2012*). Owing to the variation in sequence depths between samples, all samples were normalized to the lowest depth by subsampling (6000 read/sample) and sequences that were present in fewer than two samples (doubletons) were discarded from the final table (*Beleneva and Zhukova, 2009*). Sequences were submitted under the accession number SRP131209 in the Sequence Read Archive (SRA).

## Comparison of gut communities

Richness (number of observed species) and Shannon α-diversity were calculated using QIIME. To assess β-diversity, cluster analyses exploring the similarities between the compositions of gut communities from different samples were performed using Bray–Curtis similarity, unless otherwise stated. Adonis implementation of Permanova (*Anderson, 2001*) (non-parametric permutational multivariate analysis of variance) was used for comparison between groups. Distances between the different groups were calculated using the QIIME script 'make_distance_plots.py' and their significance was assessed using a non-parametric t-test with Monte Carlo simulation. Moreover, to examine taxonomical composition statistically and to identify taxa that were associated with different tolerance groups and temperature conditions, we applied the Dufrene–Legendre Indicator Species Analysis (*Dufrêne and Legendre, 1997*) (function *indval* with 1000 permutations, package 'labdsv' in R). This analysis calculates the indicator value (fidelity and relative abundance) of species in clusters or types, given by the following formula:

$$IndVal_{ij} = Specificity_{ij} \times Fidelity_{ij} \times 100$$

where $IndVal_{ij}$ is the indicator value of species '$i$' in relation to site type '$j$'; $Specificity_{ij}$ is the proportion of sites of type '$j$' with species '$i$', and $Fidelity_{ij}$ is the proportion of the number of individuals (abundance) of species '$i$' that are in site type '$j$'.

Core communities were identified as the most resilient microbes present in >50% of the individuals at either 24°C or 12°C. To measure the absolute abundance of the two most prevalent core microbes (*Pseudomonas veronii* and *Janthinobacterium lividum*) and to validate our sequencing

data, we designed specific primers to amplify their copy numbers using the Primer-BLAST tool (*Ye et al., 2012*). After validation of the primers, real-time PCR was performed using the specific primers (*P. veronii*, Forward 5′-CCGCGGTAATACAGAGGGTG-3′ and Reverse 5′-ACCCTCTACCATAC TCTAGTCAGT-3′; *J. lividum*, Forward 5′- CGCAGGCGGTTTTGTAAGTC-3′ and Reverse 5′- GTCAA TCTTGACCCAGGGGG-3′). The fold change in abundance of each microbe was calculated relative to that in the warm conditions as the control.

## Functional profile of the gut microbiome

To predict the functional content of the gut microbiome originating from the different groups, we used the PICRUSt tool (*Langille et al., 2013*). PICRUSt (http://picrust.github.com/picrust/) is a software package designed to infer metagenome functional content from 16S metagenomic data. The paired-end merged 16S sequences were used for closed-reference OTU picking using QIIME, and the resulting OTU table was then fed into PICRUSt and functional predictions assigned to KEGG pathways (*Kanehisa and Goto, 2000*) were made according to the metagenome inference workflow described by the developers. PICRUSt results were normalized, and then analyzed using the LEfSe tool (*Segata et al., 2011*). LEfSe is an algorithm for high-dimensional biomarker discovery and explanation that identifies genomic features (i.e., taxa) characterizing the differences between two or more biological conditions. It uses statistical significance and biological relevance to identify differentially abundant features, using the non-parametric factorial Kruskal–Wallis rank-sum test and the Wilcoxon rank-sum pairwise test to compare biologically significant categories. Then, it uses LDA to estimate the effect size of each differentially abundant feature and to perform dimension reduction to assess whether the differences are consistent with the expected biological behavior. We thus reanalyzed our sequences using the QIIME closed reference protocol against the 97% similarity to the GreenGenes database. The nonparametric t-test (using Monte Carlo simulation) was used to compare differently enriched pathways between groups. A heat map was created using the R package 'heatmap3' function. The non-relevant categories, that is human diseases and functions that are exclusive to eukaryotic organisms, were not presented in the heat map.

## Transcriptome analysis

Total RNA from the liver of selected individuals was extracted using TRIzol reagent (Invitrogen, California, USA), according to the manufacturer's instructions and treated with DNAse (TURBO-DNase, Ambion). Selection of the individuals was based on their microbiome composition (principal coordinates analysis using Bray–Curtis metric to select for individuals with discriminating microbiomes; *Figure 4—figure supplement 3*). RNA concentration and quality were determined in each sample and samples were equimolarly diluted and sent to the Technion Genome Center (The Technion, Haifa, Israel) for library preparation and single-end sequencing using Illumina HiSeq. The resultant raw short reads were subjected to a filtering and cleaning procedure as follows: the SortMeRNA tool (*Kopylova et al., 2012*) was used to filter out rRNA, and then the FASTX Toolkit (*Gordon and Hannon, 2010*) (version 0.0.13.2) was used for (a) trimming read-end nucleotides with quality scores <30 using 'fastq_quality_trimmer' and (b) removing reads with less than 70% base pairs with quality score $\leq$30 using 'fastq_quality_filter'. The Bowtie2 version 2.1 alignment tool was used to map cleaned reads onto the reference transcriptome of *Oreochromis niloticus* extracted from the NCBI database (ftp://ftp.ncbi.nlm.nih.gov/genomes/Oreochromis_niloticus/; ASM185804v2) (*Supplementary file 1*, Table S7). Then, transcript quantification was performed using the expectation-maximization method (RSEM), by estimating maximum likelihood expression levels (*Li and Dewey, 2011*). Differential expression analysis was performed with the R package 'DESeq2' (*Love et al., 2014*) and transcripts that were more than 2-fold differentially expressed with false discovery rate (FDR)-corrected p < 0.05 (*Benjamini and Hochberg, 1995*) were considered. The reference transcriptome sequences of *O. niloticus* were searched against the *Danio rerio* transcriptome reference (from the NCBI database) for homology identification. The results were further analyzed using the Database for Annotation, Visualization and Integrated Discovery (DAVID) (*Huang et al., 2009*) to determine the Gene Ontology (GO) and KEGG pathways. Last, to examine host-associated gene expression patterns statistically and to identify genes that are associated with different tolerance groups, we applied the Dufrene–Legendre Indicator Species Analysis (*Dufrene and Legendre (19977)*; function *indval* with 1000 permutations, package 'labdsv' in R), as described in the

'Comparison of gut communities' section. Transcriptome raw sequences are submitted under the accession number SRP164378 (PRJNA419688) in the SRA.

## Statistical analysis

Non-parametric tests (Wilcoxon test) and linear mixed-effect models (*nlme* R package [*Bates and Pinheiro, 1998*]) were used to assess α-diversity, while Adonis implementation of Permanova (vegan R package [*Anderson, 2001*]) was used for comparisons between groups in the analysis of β-diversity using the Bray–Curtis distance matrix. LEfSe (*Segata et al., 2011*) was used to estimate the effect size of each differentially abundant PICRUSt feature and to perform dimension reduction using the Galaxy online tool (http://huttenhower.sph.harvard.edu/galaxy/). CCA (*Butts, 2012*), was performed between the matrices of the microbiome composition table (OTUs) and the transcriptomic data using principal component analysis. The total fraction of OTU variance accounting for each component by the transcriptomic analysis (expression) variables, through all canonical variates, was calculated. For that, the first four principal components (PCs) from the OTU table were extracted (R *prcomp*). In addition, the first four expression matrix PCs were extracted using R *prcomp*. The actual value of the total variance was then compared to that of 1000 random permutations, in which the order of the host line PCs was shuffled.

## Acknowledgements

We would like to thank the editors and the reviewers for their valuable comments that helped to improve our manuscript. The research described here was supported by grants from the European Research Council under the European Union's Horizon 2020 research and innovation program (grant agreement no. 640384), the Israel Science Foundation (grant no. 1313/13), and the Chief Scientist of the Ministry of Agriculture and Rural Development (grant no. 863–0045).

## Additional information

### Funding

| Funder | Grant reference number | Author |
| --- | --- | --- |
| European Research Council | Grant 640384 | Fotini Kokou<br>Goor Sasson<br>Tali Nitzan<br>Adi Doron-Faigenboim<br>Sheenan Harpaz<br>Avner Cnaani<br>Itzhak Mizrahi |
| Israel Science Foundation | Grant number 1313/13 | Fotini Kokou<br>Goor Sasson<br>Tali Nitzan<br>Adi Doron-Faigenboim<br>Sheenan Harpaz<br>Avner Cnaani<br>Itzhak Mizrahi |
| Ministry of Agriculture and Rural Development | Grant number 863-0045 | Fotini Kokou<br>Goor Sasson<br>Tali Nitzan<br>Adi Doron-Faigenboim<br>Sheenan Harpaz<br>Avner Cnaani<br>Itzhak Mizrahi |

The funders had no role in study design, data collection and interpretation, or the decision to submit the work for publication.

### Author contributions

Fotini Kokou, Conceptualization, Data curation, Formal analysis, Validation, Visualization, Methodology, Writing—original draft; Goor Sasson, Data curation, Formal analysis; Tali Nitzan, Data curation; Adi Doron-Faigenboim, Data curation, Software; Sheenan Harpaz, Conceptualization, Project

administration; Avner Cnaani, Itzhak Mizrahi, Conceptualization, Resources, Supervision, Funding acquisition, Investigation, Writing—original draft, Project administration

## Author ORCIDs
Fotini Kokou [iD] http://orcid.org/0000-0002-3675-3835
Itzhak Mizrahi [iD] http://orcid.org/0000-0001-6636-8818

## Ethics
Animal experimentation: This study was approved by the Agricultural Research Organization Committee for Ethics in Using Experimental Animals and was carried out in compliance with the current laws governing biological research in Israel (Approval number: 146/09IL).

## Decision letter and Author response
Decision letter https://doi.org/10.7554/eLife.36398.028
Author response https://doi.org/10.7554/eLife.36398.029

## Additional files

### Supplementary files
• Supplementary file 1. Supplementary tables.
DOI: https://doi.org/10.7554/eLife.36398.023

• Transparent reporting form
DOI: https://doi.org/10.7554/eLife.36398.024

### Data availability
Data has been deposited in the SRA under accession code SRP131209.

The following dataset was generated:

| Author(s) | Year | Dataset title | Dataset URL | Database and Identifier |
|---|---|---|---|---|
| Fotini Kokou, Goor Sasson, Itzhak Mizrahi | 2018 | Blue tilapia gut microbiome | https://www.ncbi.nlm.nih.gov/sra/SRP131209 | NCBI Sequence Read Archive, SRP131209 |

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
