## [Decision Letter]

Thank you for submitting your article "Support for the hologenome theory as host adaptive capacity shapes and modulates its microbiome response to temperature" for consideration by *eLife*. Your article has been reviewed by three peer reviewers, and the evaluation has been overseen by a Reviewing Editor and Wendy Garrett as the Senior Editor. The reviewers have opted to remain anonymous.

The reviewers have discussed the reviews with one another and the Reviewing Editor has drafted this decision to help you prepare a revised submission.

Summary:

This manuscript by Kokou et al. proposes that Tilapia fish and its microbiome evolve as a unit of selection, based on measured correlations between host genetic makeup and microbiome responses to temperature. The consensus opinion is that, at best, the results claimed by the authors show a relationship (i.e. correlation) between host resistance and microbiome "resilience". This is very interesting, but far from showing support for the hologenome theory. Therefore, the paper needs to be completely rewritten to focus on the experiments actually done and omit the tie-in to the hologenome theory, or transgenerational experiments need to be performed.

Essential revisions:

The authors have measured a microbiome response to temperature within one generation rather than transgenerational changes in the microbiome that would more directly support the hologenome concept of evolution. More generally, it is important to note that the hologenome concept of evolution is an eco-evolutionary hypothesis, while the terms holobiont are hologenome are structural definitions used to define the host-microbiome ecosystem and their genomes.

Although multilevel selection is an interesting concept that could apply to host-microbiome associations, extraordinary claims demand extraordinary evidence, and this paper falls short of providing such evidence. In particular, the study shows no data on evolution, only ecological changes in microbiome composition correlated to host genetics. Multi-level selection or coevolution would require transgenerational experiments to establish, not possible in the two month revision window. Given that this paper is not about multilevel selection, presenting that as the main point of this study appears misleading. This is why the paper should be rewritten to say that it is "consistent with the hologenome theory", with this being either a minor component or omitted entirely, or new data supporting evolutionary adaptation effects need to be provided through new experiments.

Focusing on the main data of the paper, which describes the correlation between host resistance to stress and microbiome composition and response, the main result is expressed in this sentence:

"Interestingly, when fish from both groups were kept under their optimal temperature conditions of 24°C, the sensitive fish exhibited higher microbial diversity (Shannon H') than the resistant ones (Supplementary Figure 2; *P* = 0.04), suggesting that host adaptive capacity is involved in shaping microbiome composition."

What this shows is that the host genetics and physiology (not its adaptive capacity) conditions the composition of the microbiome, at least in this broad statistical sense. This is a strong result which the authors would expand more to provide some insight into processes and or mechanisms.

Other claims, however, are only weakly supported if at all. Part of the problem is the use of statistical descriptors that do not capture the multiple dimensions of the data. The authors choose to do the analysis of microbiome composition and susceptibility to temperature at the level of non-informative summary statistics, like diversity indexes. These numbers hide a lot of information about what is going on in the data, and the authors derive conclusions from simple statistical comparisons of diversity distributions that are not well supported.

For instance, Figure 2A shows that there are many individuals with similar diversity at 24 and 12°C, but that the 24°C control has a tail of high diversity microbiomes that drives the statistically significant difference between treatment and control. What would happen if we stratify the data by diversity in order to compare individuals with similar diversity indexes in the two families? Would all low-diversity microbiomes behave the same regardless of family? In other words, is the effect we see simply a *population* average driven by a few more variable individuals, or can it be pinned down to features of the individual microbiome composition? This is an example of how the analysis presented by the authors hides a lot of important information that prevents the reader from getting away with more than a few headlines and no insight into the actual processes.

One of the main points of the paper is that the microbiome of resistant individuals is more "resilient" to changes in temperature. However, if diversity is already low in the resistant family and if temperature stress decreases diversity, it stands to reason that diversity will drop less in the resistant family compared to the sensitive one, whose microbiomes still have more microbial taxa "to spare". In other words, drops in diversity may saturate, either because there is a biological lower bound to it or because the index has a non-linear behavior. To assess "resilience" it would be better to work with the set of taxa that are present in all microbiomes (the core), renormalize the data and study fold changes. Again, the problem described in this paragraph stems from the attempt to study changes in high dimensional species composition profiles with single numbers like richness or Shannon diversity.

Unfortunately, the taxonomic analysis, which could deal with part of the problem, is done only at the level of shallow clades. For instance, most organisms in Figure 2B are proteobacteria. As the authors must know, this is an incredibly diverse set of microbes. Which proteobacteria? The authors should describe fold-changes in taxa at various taxonomic levels, starting with exact sequence variants.

Finally, the functional analysis is strongly biased by taxonomic changes. For example, the statement that the "host selects for taxa with a specific functional landscape that is better suited to cold exposure" is a bold claim lacking supporting evidence, What is shown is simply an enrichment of functions that are more frequently found in proteobacteria. The type of functional categories found have no relationship to anything related to adaptation to low temperatures. Incorporation of David Sloan Wilson's concepts of "selection for" vs. "selection of" traits might be helpful here.

In sum, the finding that the microbiome composition is related to the genetic makeup of the host is interesting, but the authors should dig deeper into processes and, if possible, mechanisms. A better statistical analysis that goes beyond simple summary statistics like average richness and diversity and focuses on specific sequence variants and their relationship to the resistance/sensitivity of individual hosts could make the paper much better. The authors should refrain from making grandiose claims about multilevel selection, microbiome resilience, etc., and simply focus on providing a more in-depth analysis of their data.

---

## [Author Response]

Essential revisions:The authors have measured a microbiome response to temperature within one generation rather than transgenerational changes in the microbiome that would more directly support the hologenome concept of evolution. More generally, it is important to note that the hologenome concept of evolution is an eco-evolutionary hypothesis, while the terms holobiont are hologenome are structural definitions used to define the host-microbiome ecosystem and their genomes.Although multilevel selection is an interesting concept that could apply to host-microbiome associations, extraordinary claims demand extraordinary evidence, and this paper falls short of providing such evidence. In particular, the study shows no data on evolution, only ecological changes in microbiome composition correlated to host genetics. Multi-level selection or coevolution would require transgenerational experiments to establish, not possible in the two month revision window. Given that this paper is not about multilevel selection, presenting that as the main point of this study appears misleading. This is why the paper should be rewritten to say that it is "consistent with the hologenome theory", with this being either a minor component or omitted entirely, or new data supporting evolutionary adaptation effects need to be provided through new experiments.

The manuscript was adjusted to focus on the experiments done and new analysis was also added according to the reviewers’ suggestions (see below). In addition, the tie-in to the hologenome concept was omitted from the title and moderated to be more conceptual.

Focusing on the main data of the paper, which describes the correlation between host resistance to stress and microbiome composition and response, the main result is expressed in this sentence:"Interestingly, when fish from both groups were kept under their optimal temperature conditions of 24 °C, the sensitive fish exhibited higher microbial diversity (Shannon H') than the resistant ones (Supplementary Figure 2; P = 0.04), suggesting that host adaptive capacity is involved in shaping microbiome composition."What this shows is that the host genetics and physiology (not its adaptive capacity) conditions the composition of the microbiome, at least in this broad statistical sense. This is a strong result which the authors would expand more to provide some insight into processes and or mechanisms.

Thank you for this comment. According to your suggestion, we further analysed our data to highlight how host genetics/cold tolerance and physiology condition the microbiome. We first performed a similarity test to compare the overall microbiome composition between the cold-resistant and cold-sensitive hosts using Bray-Curtis metric (non-parametric t-test with 1000 Monte Carlo permutations) at the control conditions (24^o^C) to see potential differences stemming from the genetic selection. We found that the microbiome response within the resistant individuals was more similar compared to the sensitive (Figure 2Ci). We further aimed to get a better insight into the microbiome composition of these host groups, thus we compared the microbiome composition between the cold-resistant and cold-sensitive hosts under any temperature condition; more specifically, we asked whether microbiomes of similar (Shannon H’) diversity were also sharing the same microbiome compositions. To this end, we stratified our data and examined individuals from each host group with similar diversity (see in the next comments for more details). This analysis showed that the microbiomes clustered significantly according to host group/tolerance under any temperature conditions (Figure 2—figure supplement 6), thus suggesting that host genetic background (genetic selection for cold tolerance) has a strong effect on shaping the microbiome composition. We next explored microbial taxa associated to genetic background effects and applied the indicator species analysis (this measure identifies habitat-associated species based on their fidelity and relative abundance in different environments; see Materials and methods). We found different microbes to be associated with the cold-resistant and cold-sensitive fish (Figure 2D), namely *Cetobacterium somerae* species were mostly enriched in the resistant, while a higher diversity of taxa such as *Streptococcus luteciae, Bacteroides fragilis, Prevotella sp., Clostridiaceae* and *Succinivibrionaceae* species were enriched in the sensitive (Table S5 in Supplementary file 1). Interestingly, in the cold conditions we found an enrichment of *Vibrionales* species in the resistant fish, which have been shown to include several psychrophilic taxa or taxa with adaptations for cold tolerance, thus indicating that cold acclimation of the host may be selecting for specific microbes, with some of them potentially carrying fitness traits for cold tolerance (see below for more information).

These results and insights were obtained after we re-analysed all our data using DADA2 (Callahan et al., 2016) as suggested by the reviewers (see below and in the Materials and methods section). Our overall results and figures, generated using the new data at the exact sequence variant level (ESV) originating from DADA2, show the same patterns and conclusions (Figures 2, 3, 4; See also supplementary figures).

Other claims, however, are only weakly supported if at all. Part of the problem is the use of statistical descriptors that do not capture the multiple dimensions of the data. The authors choose to do the analysis of microbiome composition and susceptibility to temperature at the level of non-informative summary statistics, like diversity indexes. These numbers hide a lot of information about what is going on in the data, and the authors derive conclusions from simple statistical comparisons of diversity distributions that are not well supported.For instance, Figure 2A shows that there are many individuals with similar diversity at 24 and 12 °C, but that the 24 °C control has a tail of high diversity microbiomes that drives the statistically significant difference between treatment and control. What would happen if we stratify the data by diversity in order to compare individuals with similar diversity indexes in the two families? Would all low-diversity microbiomes behave the same regardless of family? In other words, is the effect we see simply a population average driven by a few more variable individuals, or can it be pinned down to features of the individual microbiome composition? This is an example of how the analysis presented by the authors hides a lot of important information that prevents the reader from getting away with more than a few headlines and no insight into the actual processes.

This is a valid point by the reviewers. We thank you for this comment as it allowed us to better present our results and understand our observations.

Following this suggestion, we stratified our data by diversity to compare individuals with similar diversity indices. We selected the 10 most extreme values from both temperatures (lowest and highest Shannon H’ diversity; Table S2 in Supplementary file 1) and we performed multivariate and clustering analysis (Two-way Permanova, Tables S3 and S4 in Supplementary file 1; Principal Coordinate Analysis, Figure 2—figure supplement 6). Our results show that, even when we select individuals with similar diversity indices, either high or low, their posterior gut microbiomes cluster significantly according to temperature and host genetics (tolerance) group (Two-way Permanova in Tables S3 and S4 in Supplementary file 1; Figure 2—figure supplement 6).

In addition, in our attempt to identify specific features that contribute to these changes, we performed the Dufrene-Legendre Indicator Species Analysis (Dufrene and Legendre, 1997; function *indval* with 1000 permutations, package "*labdsv*" in R), as described now in the Materials and methods section (subsection “Comparison of gut communities”). This analysis calculates the indicator value of a species in clusters or types, given by the following formula:

IndVal*ij* = Specificity*ij* × Fidelity*ij* × 100

where IndVal*i j* is the Indicator Value of an '*i*' species in relation to a '*j*' type of site; Specificity *ij* is the proportion of sites of type '*j*' with species '*i*', and Fidelity *ij* is the proportion of the number of individuals (abundance) of species '*i*' that are in a '*j*' type of site.

This analysis showed several ESV features to be host genetics (tolerance)-associated in the low and high diversity microbiomes, as shown in Table S5 in Supplementary file 1. The resistant fish with low diversity microbiomes were more enriched with species of the *Vibrionales* order, which include taxa that are cold-resistant (Raymond-Bouchard, Whyte, 2017; Townsley et al., 2016), while in the warm temperature, we saw in the high diversity microbiomes an enrichment of *Cetobacterium somerae,* which has been previously reported in the tilapia gut environment (Tsuchiya et al., 2008; Giatsis et al., 2015; Haygood, Jha, 2016). In the high diversity microbiome of the sensitive group, we observed enrichment of microbes within the *Succinivibrionaceae* and *Clostridiaceae* family (unknown species), with the latter being previously reported in several warm water fish species including tilapia, along with *Streptococcus luteciae, Bacteroides fragilis* and *Prevotella* species.

Looking at host genetics-associated posterior gut microbiome at different taxonomic levels (Figure 2D; Figure 2—figure supplement 8; Table S6 in Supplementary file 1), similarly we found significant enrichment of the *Vibrionales* order in the cold-resistant fish, while, as mentioned above, a higher microbial order diversity such as *Clostridiales, Methanobacteriales* and *Actinomycetales* in the sensitive fish. Altogether, our results suggest that host genetic background shapes the microbiome composition, with some members that potentially carry fitness traits for cold tolerance (i.e. species from the *Vibrionales* order).

One of the main points of the paper is that the microbiome of resistant individuals is more "resilient" to changes in temperature. However, if diversity is already low in the resistant family and if temperature stress decreases diversity, it stands to reason that diversity will drop less in the resistant family compared to the sensitive one, whose microbiomes still have more microbial taxa "to spare". In other words, drops in diversity may saturate, either because there is a biological lower bound to it or because the index has a non-linear behavior. To assess "resilience" it would be better to work with the set of taxa that are present in all microbiomes (the core), renormalize the data and study fold changes. Again, the problem described in this paragraph stems from the attempt to study changes in high dimensional species composition profiles with single numbers like richness or Shannon diversity.

Following your suggestion, we assessed the resilience of microbiomes that were present in >50% of the individuals in both temperatures (the core microbiome). We renormalized the data (re-calculated their relative abundance within the core microbiome) and examined the fold changes between the two temperatures within each tolerant group (Figure 2E). Overall, we observed that the fold change of these core microbes was significantly less affected by the temperature change in the resistant fish compared to the sensitive ones (Figure 2Eii). In addition, we designed specific primers in order to measure the absolute numbers for the most abundant and prevalent core taxa (>95% of the individuals; *Pseudomonas veronii* and *Janthinobacterium lividum*) and quantified their counts within the samples using quantitative PCR (See in the Materials and methodssubsection “Comparison of gut communities”). Our results agree with the sequencing data, showing a significantly less pronounced effect on the absolute abundances of the core microbes in the gut of resistant fish. These results, from both re-normalization of the core microbes’ abundance and the absolute numbers of some of these core taxa using real-time PCR, support our findings by showing that the changes in the microbiome composition are significantly less pronounced in the resistant fish compared to the sensitive ones (Figure 2E; *P* < 0.05), thus supporting our hypothesis that host thermal tolerance modulates microbiome response to temperature changes.

Unfortunately, the taxonomic analysis, which could deal with part of the problem, is done only at the level of shallow clades. For instance, most organisms in Figure 2B are proteobacteria. As the authors must know, this is an incredibly diverse set of microbes. Which proteobacteria? The authors should describe fold-changes in taxa at various taxonomic levels, starting with exact sequence variants.

As suggested by the reviewers, we presented the taxonomic analysis of the microbiome at each taxonomic level (Figure 2B; Figure 2—figure supplement 4), as well as calculated the fold change from control to cold conditions for each group presented in Figure 2—figure supplement 5. In addition, as described above, we performed indicator species analysis for each of the taxonomic levels that showed several host-associated taxonomic groups.

As suggested we re-analysed all our data using DADA2 (Callahan et al., 2016) in order to correct any potential sequencing errors and presented our data at the Exact Sequence Variant (ESV) level (100% sequence identity dataset). In addition to using this tool, we took even more stringent approach in which we eliminated sequences present in less than two samples (doubletons) from the 100% sequence ID data sets. This overall analysis is described in the Materials and methods section (subsection “Sequencing of gut microbiome”). Essentially, our overall results and figures, generated using the new ESV data originating from DADA2, show the same patterns and conclusions (Figures 2, 3, 4; See also supplementary figures).

As presented in Figure 2A and B, the sensitive animals had a higher diversity of microbial orders compared to the resistant ones, such as species from the families *Clostridiaceae, Succinivibrionaceae* and from the genera *Prevotella* and *Streptococcus* (Figure 2—figure supplement 7-8). These species consistently existed in most of the individuals and show significant association for the sensitive (genetic) group as it emerges from the *indval* analysis (Figure 2D; 6 in Supplementary file 1 and Figure 2—figure supplement 8). Similarly, we found significant association to the resistant hosts of taxa originating mostly from the *Vibrionales* order (Figure 2D; Table S6 in Supplementary file 1 and Figure 2—figure supplement 8). Interestingly, taxa from this order has been reported as resistant to low temperatures or isolated from deep sea waters (Bouchard and Whyte, 2017). More specifically, *Vibrio cholerae*, a naturally occurring species in the freshwater fish gut (Haygood and Jha, 2018), which we found enriched in the resistant fish gut after cold exposure (Figure 2—figure supplement 5F), has been reported to be resistant to low temperatures through adaptation in its physiology (Townsley et al., 2016). Such results support our claims, that host genetic background, but also cold acclimation phenotype is associated with specific microbiome composition, with some of them potentially carrying fitness traits for cold tolerance.

Finally, the functional analysis is strongly biased by taxonomic changes. For example, the statement that the "host selects for taxa with a specific functional landscape that is better suited to cold exposure" is a bold claim lacking supporting evidence, What is shown is simply an enrichment of functions that are more frequently found in proteobacteria. The type of functional categories found have no relationship to anything related to adaptation to low temperatures. Incorporation of David Sloan Wilson's concepts of "selection for" vs. "selection of" traits might be helpful here.

Our claim of ‘host selects for taxa with a specific functional landscape that is better suited to cold exposure’ was removed from the text. We additionally discuss and cite David Sloan Wilson's concepts considering our results in the discussion. The taxonomic groups that we found to be enriched in the gut are associated with the hosts and the environmental changes; this also results in the enrichment of functions associated with these taxa, with this not necessarily being connected to cold tolerance. Moreover, to investigate whether there are shared responses between the host and the microbiome after cold exposure that could suggest cold acclimation or cold stress response, we compared the significantly affected KEGG orthologs within the microbiome (Figure 3A) to that of the host transcriptome. Interestingly, we found several functions that were not only shared, but also similarly affected (Figure 3C; Figure 3—figure supplement 1B) between the host and the microbiome. Some of the functions affected were previously reported as key cell metabolism cold-induced adaptations in mice (i.e. down-regulation of PPAR signalling, cytochome 450; Shore et al., 2013) and thus suggesting a potential conserved response to low temperatures.

In sum, the finding that the microbiome composition is related to the genetic makeup of the host is interesting, but the authors should dig deeper into processes and, if possible, mechanisms. A better statistical analysis that goes beyond simple summary statistics like average richness and diversity and focuses on specific sequence variants and their relationship to the resistance/sensitivity of individual hosts could make the paper much better. The authors should refrain from making grandiose claims about multilevel selection, microbiome resilience, etc., and simply focus on providing a more in-depth analysis of their data.

Thank you for this remark. As mentioned above, we now re-analysed our data obtaining a more in-depth taxonomic information at the Exact Sequence Variant level using DADA2. Nonetheless, we further developed our insights on the microbial composition changes due to the host genetic makeup and thermal tolerance.

We have shown that when fish from both groups were kept under their optimal temperature conditions of 24^o^C, the sensitive fish exhibited higher microbial diversity (Shannon H’) than the resistant ones (Figure 2Aii), as well as a higher individual variability. Indeed, when we compared the microbial community structure (β-diversity) of the two groups at these optimal conditions, we found that cold-resistant fish had a higher within-group microbiome similarity compared to the sensitive fish (Figure 2Ci), potentially due to the selection process that resistant fish have been subjected to regarding the cold resistant trait and thus showing a host control on the microbiome composition.

We further aimed to get a better insight into the microbiome composition of these host groups. To do so, we first compared the microbiome composition between the cold-resistant and cold sensitive hosts under any temperature condition; more specifically, we asked whether microbiomes of similar (Shannon H’) diversity are also sharing the same microbiome compositions. We stratified our data and examined individuals from each host group with similar Shannon H’ diversity (individuals from each group with the 10 highest or 10 lowest Shannon H’ Diversity values; Table S2 in Supplementary file 1). We found that the individuals from both high- and low-diversity microbiomes (Principal Coordinate Analysis; Figure 2—figure supplement 6) clustered significantly according to the host group (Permanova analysis, Tables S3 and S4 in Supplementary file 1), suggesting that host genetic background (genetic selection for cold tolerance) has a strong effect on shaping the microbiome composition.

Our next step was to explore microbial taxa associated to genetic background effects and applied the indicator species analysis (this measure identifies habitat-associated species based on their fidelity and relative abundance in different environments; see above and Materials and methods). This analysis revealed several microbial taxa to be significantly associated with host genetic background, previously described as naturally occurring strains in the tilapia gut (Giatsis et al., 2015; Haygood and Jha, 2018) (Tables S5 and S6 in Supplementary file 1; Figure 2—figure supplement 7-8). Namely, microbial species such as *Cetobacterium somerae* (Tsuchiya et al., 2008) and species from the *Vibrionales* order (i.e. *Vibrio cholerae*; Figure 2—figure supplement 5F), described also as psychrophiles or cold-resistant microbes surviving in different environments (Raymond-Bouchard, Whyte, 2017; Townsley et al., 2016) were enriched in the resistant fish gut, while a higher diversity of taxa was enriched in the sensitive fish gut including *Prevotella sp., Streptococcus luteciae*, and *Bacteroidetes* species, as well as species from the *Christensenellaceae, Succinivibrionaceae* and *Clostridiaceae* families. Overall, the gut of the sensitive fish was more diverse in microbial taxa and more variable between individuals during the warm conditions compared to the resistant fish, which could potentially stem from the fact that the resistant fish have been subjected to a selection process towards the cold resistant trait, while sensitive ones still contain higher genetic variability, however this still remains to be evaluated in future experiments. Altogether, our results suggest that host genetic background shapes the microbiome composition by selecting for specific microbes, with some of them potentially carrying fitness traits for cold tolerance.

Finally, as host fish were specifically selected according to their thermal tolerance, and our findings indicated that temperature and host genetic background affect the fish gut microbiome, we further asked whether the microbiome's response to temperature stress is also affected by its host's tolerance. When exposed to the cold temperature of 12^o^C, both resistant and sensitive hosts exhibited a decrease in their microbiome diversity and richness (Figure 2A; Figure 2—figure supplement 3), as well as changes in its composition (Figure 2B; Figure 2—figure supplement 4-5). However, when we compared the β-diversity within each of these host groups in relation to temperature change, we found that after exposure to the stressful low-temperature conditions, the microbiomes of cold-resistant families were less affected than those of sensitive ones (Figure 2Cii). More specifically, when we compared the similarity of the microbial compositions between 24^o^C and 12^o^C, we found it to be significantly higher in the resistant families than in the sensitive ones, showing a stronger effect of temperature change in the sensitive fish (Figure 2Cii). We next asked whether this differential response to temperature of the microbiome in the two host groups also applies to microbes that are shared between the two host groups. To this end, we assessed the microbiome resilience in 11 selected taxa that were present in >50% of the individuals from each temperature (core microbiome). After renormalizing the core microbes abundance (expressed as percentage within the core microbiome community; Figure 2E), we evaluated the fold changes in their abundance from 24^o^C to 12^o^C in the cold-resistant and cold-sensitive fish. Similar to the overall microbiome response (Figure 2Cii), the fold changes in the relative abundance of the core microbiome in the resistant fish was significantly lower that the sensitive ones (Figure 2Eii). In addition, we further designed specific primers in order to measure the absolute copy numbers for the two most abundant and prevalent core taxa within both host groups and quantified their abundance using quantitative PCR (see Materials and methods). Our results were in agreement with the sequencing data (Figure 2Ei, ii), showing a significantly less pronounced effect of the temperature on the abundance of the core microbes in the gut of the resistant fish compared to the sensitive ones (Figure 2Eiii). This means that the sensitivity of the shared-core microbes to cold temperatures was higher in the sensitive fish and thus supporting our hypothesis that host thermal tolerance modulates its microbiome response to temperature changes.